# The Effect of Quercetin on Non-Alcoholic Fatty Liver Disease (NAFLD) and the Role of Beclin1, P62, and LC3: An Experimental Study

**DOI:** 10.3390/nu16244282

**Published:** 2024-12-11

**Authors:** Ioannis Katsaros, Maria Sotiropoulou, Michail Vailas, Fotini Papachristou, Paraskevi Papakyriakopoulou, Marirena Grigoriou, Nikolaos Kostomitsopoulos, Alexandra Giatromanolaki, Georgia Valsami, Alexandra Tsaroucha, Dimitrios Schizas

**Affiliations:** 1First Department of Surgery, National and Kapodistrian University of Athens, Laikon General Hospital, 11527 Athens, Greece; marosotiropoulou@gmail.com (M.S.); mike_vailas@yahoo.com (M.V.); schizasad@gmail.com (D.S.); 2Laboratory of Experimental Surgery, Faculty of Medicine, Democritus University of Thrace, 68100 Alexandroupolis, Greece; fpapachr@med.duth.gr (F.P.); atsarouc@med.duth.gr (A.T.); 3Laboratory of Biopharmaceutics-Pharmacokinetics, Department of Pharmacy, School of Health Sciences, National and Kapodistrian University of Athens, 15774 Athens, Greece; ppapakyr@pharm.uoa.gr (P.P.); valsami@pharm.uoa.gr (G.V.); 4Laboratory of Molecular Developmental Biology & Molecular Neurobiology, Department of Molecular Biology and Genetics, Democritus University of Thrace, 68100 Alexandroupoli, Greece; mgrigor@mbg.duth.gr; 5Laboratory Animal Facility, Biomedical Research Foundation of the Academy of Athens, 11527 Athens, Greece; nkostom@bioacademy.gr; 6Department of Pathology, University Hospital of Alexandroupolis, Democritus University of Thrace, 68100 Alexandroupolis, Greece; agiatrom@med.duth.gr

**Keywords:** NAFLD, MASLD, quercetin, autophagy, Beclin1, LC3, p62, SQSTM1

## Abstract

**Background/Objectives:** Non-alcoholic fatty liver disease (NAFLD) is a major metabolic disorder with no established pharmacotherapy. Quercetin, a polyphenolic flavonoid, demonstrates potential hepatoprotective effects but has limited bioavailability. This study evaluates the impact of quercetin on NAFLD and assesses the roles of autophagy-related proteins in disease progression. **Methods:** Forty-seven male C57BL/6J mice were fed a high-fat diet (HFD) for 12 weeks to induce NAFLD, followed by quercetin treatment for 4 weeks. Mice were divided into baseline, control, and two quercetin groups, receiving low (10 mg/kg) and high (50 mg/kg) doses. Liver histology was scored using the NAFLD Activity Score (NAS). Immunohistochemistry and immunoblotting were performed to analyze autophagy markers. **Results:** Quercetin-treated groups showed significant reductions in NAS compared to controls (*p* = 0.011), mainly in steatosis and steatohepatitis. Immunohistochemistry indicated increased expression of autophagy markers LCA and p62 in quercetin groups. Western blot analysis revealed significant elevations in LC3A in the treated groups, suggesting improved autophagic activity and lipid degradation. **Conclusions:** Quercetin effectively reduces NAFLD severity and modulates autophagy-related proteins. These findings suggest that quercetin enhances autophagic flux, supporting its therapeutic potential for NAFLD. Additional research is needed to clarify the molecular mechanisms of quercetin and to determine the optimal dosing for clinical application.

## 1. Introduction

Non-alcoholic fatty liver disease (NAFLD) is a major public health disease characterized by the accumulation of fat in the liver without a history of other liver lipid accumulation causes, such as significant alcohol consumption [1,2]. The European Association for the Study of the Liver (EASL) introduced the term metabolic dysfunction-associated fatty liver disease (MAFLD) in 2020 to more accurately represent its pathophysiological basis [3]. NAFLD has a worldwide prevalence that reaches 30% and is predicted to affect more than 100 million people in the Western world by 2030 [4,5]. Sedentary lifestyles, low levels of physical activity, and imbalanced diets contribute to poor metabolic health and the high prevalence of the disease [6].

The development of NAFLD was believed to be the result of the “two-hits” pathophysiologic pathway, where liver lipid accumulation in the hepatocytes leads to inflammation and fibrosis [7,8]. Liver steatosis is the result of an imbalance in the amount of fatty acids’ input and output in the hepatocytes and is associated with metabolic disorders, such as central obesity and resistance to insulin [9]. In the setting of settled liver steatosis, the “second hit” comes from several factors, including cytokines, bacterial endotoxins, and mitochondrial dysfunction, and it leads to inflammation [10]. Nevertheless, recent studies have suggested the “multiple parallel hits” hypothesis, which refers to the concomitant action of all the abovementioned factors inducing NAFLD [11]. NAFLD’s severity ranges from non-alcoholic fatty liver (NAFL) to non-alcoholic steatohepatitis (NASH) and can progress to liver cirrhosis and hepatocellular carcinoma (HCC) [12,13]. Prompt diagnosis in the initial stages of the disease and lifestyle modifications could potentially lead to the reversal of NAFLD, whereas end stages are characterized by irreversible liver injury and warrant liver transplantation [2]. No definite therapeutic agent has yet been identified, but several natural compounds have shown promising results against NAFLD [14].

Natural polyphenols represent a diverse group of polyphenolic compounds utilized in the treatment of metabolic disorders, with demonstrated hepatoprotective properties [15]. This protective effect primarily stems from enhanced lipid breakdown in the liver and a reduction in fibrotic lipolysis, thereby mitigating oxidative stress and subsequent hepatocellular damage [16]. Quercetin (QUE), a natural polyphenolic flavonoid, exhibits anti-inflammatory, antioxidant, anti-apoptotic, and immunoprotective properties [17]. Its hepatoprotective role has already been established by several studies, but the pathophysiological mechanisms behind its effect on NAFLD are not fully understood [18].

The process of autophagy is crucial to inhibit intracellular degeneration and plays a vital role in maintaining liver homeostasis [19]. It regulates hepatocellular lipid metabolism and intracellular lipid stores, and its promotion has a hepatoprotective role [20]. Several studies have shown that hepatic autophagy is inhibited in established NAFLD, but the background pathophysiologic mechanisms have not been fully understood [21,22]. The downregulation of autophagy proteins, the decreased lysosomal proteolytic activity, and the defective autophagosome–lysosome fusion have been proposed as possible mechanisms that induce autophagy inhibition [19]. Autophagy regulators, including Beclin1, protein p62/Sequestosome 1 (SQSTM1) and Microtubule-associated protein 1A/1B-light chain 3 (LC3) play a crucial role in autophagy pathways, and their assessment will contribute towards the deeper understanding of autophagy processes in NAFLD establishment and severity [23,24,25].

NAFLD is a major global health challenge driven by metabolic dysfunction, oxidative stress, and chronic low-grade inflammation. This study aimed to assess the effect of quercetin when administered orally complexed with HP-β-CD in the form of Que-HP-β-CD lyophilized product on NAFLD and to explore the role of key autophagy-related proteins (Beclin1, p62, LC3A, and LC3B), in the disease’s progression and potential therapeutic modulation.

## 2. Materials and Methods

### 2.1. Experimental Model

This experimental study was conducted in the animal facility of the Center of Clinical, Experimental Surgery and Translational Research of the Biomedical Research Foundation of the Academy of Athens. Approval for this study was received by the Veterinary Authorities of Region of Athens, Greece (Ethical approval no. 792333/4.12.2019) and the Bioethics Committee of National and Kapodistrian University of Athens (no. 239/28.1.2020).

Forty-seven male C57BL/6J (wild type) mice were recruited and fed a high-fat diet (HFD–21% fat, 0.15% cholesterol, and 19.5% casein; Harland, Teklad, TD88137) for 12 weeks to reach NAFLD [26]. Afterward, the mice were randomly allocated into four groups: baseline group (BSL, *n* = 11), which was then euthanized under deep sleep anesthesia; control group (CONTROL, *n* = 12), which received HFD for additional 4 weeks; Que-HP-β-CD group 1 (QUE 1, *n* = 12), which received HFD and the Que-HP-β-CD lyophilized product reconstituted in water for injection (WFI) (equivalent to quercetin 10 mg/kg) orally by gavage technique, for additional 4 weeks; Que-HP-β-CD quercetin group 2 (QUE 2, *n* = 12), which received HFD and the Que-HP-β-CD lyophilized product reconstituted in water for injection (WFI) (equivalent to quercetin 50 mg/kg) orally by gavage technique, for additional 4 weeks. At the end of the experimental period, all mice were euthanized under deep anesthesia. Histopathological assessment of liver samples was conducted using the NAFLD Activity Score (NAS). Furthermore, Beclin 1, P62 and LC3 expressions were also assessed using immunochemistry. A Western blot analysis followed to assess the LC3A, LC3B, and p62 levels in subcellular fractions.

### 2.2. Chemicals and Reagents

Quercetin (MW 302.24 g/mol) and hydroxypropyl-β-cyclodextrin (HP-β-CD; MW 1460 g/mol) were purchased from Sigma-Aldrich (St. Louis, MO, USA) and Ashland (Covington, KY, USA), respectively. HPLC-grade water from Fisher Scientific was used in all preparations.

### 2.3. Preparation of QUE-HP-β-CD Lyophilized Product

Lyophilized powder of Que-HP-β-CD was prepared by freeze-drying aqueous solutions of Que-HP-β-CD, following the method described by Manta et al. [27], with molar ratios of 1:2. Briefly, 4.8 g of HP-β-CD were placed in a 600 mL beaker and suspended in 500 mL of water. Then, 500 mg of QUE was added under continuous stirring and protected from light due to the molecule’s photosensitivity. Small amounts of 6% (*v*/*v*) ammonium hydroxide were gradually added until complete dissolution of QUE while maintaining the pH at approximately 9.0–9.5. The resulting solution was transferred into round trays, frozen at −73 °C, and freeze-dried using a Vacuum Freeze Dryer [BK-FD10T, Biobase Biodustry (Shandong) CO., LTD (China)]. The QUE content in the lyophilized powders was quantified using high-pressure liquid chromatography (HPLC) following the method described by Papakyriakopoulou et al. [28]. The amount of QUE in the lyophilized product was found to be 7.3 ± 0.15 % (*w*/*w*). The formation of the Que-HP-β-CD has been confirmed in our previous study applying Differential Scanning Calorimetry (DSC) [27] and characterized using XRD and SEM techniques [28]. The produced lyophilized powder is in an amorphous state, in the form of flake-like particles randomly distributed in irregular shapes of various sizes [28]. Furthermore, the prepared Que-HP-β-CD lyophilized product was found to have 30–45 times higher aqueous solubility and 3–6 times higher dissolution compared to the pure Que at pH ranging from 1.2 to 6.8 at 37 °C [29].

### 2.4. NAFLD Activity Score

The “NAFLD Activity Score–NAS” was developed in 2005 by the NASH Clinical Research Network to assess and quantify the disease severity [30]. It represents the sum of three components: steatosis (score 0–3), lobular inflammation (score 0–3), and hepatocyte ballooning (score 0–2). The minimum criterion for NAFLD is a steatosis greater than 5% [30]. The diagnosis of NASH is associated with a NAS equal to or greater than 5, while a score of 3 or 4 is considered inconclusive [30].

### 2.5. Immunohistochemistry

Paraffin sections of the livers were collected after the necessary preparation, and we applied immunohistochemistry methods for the detection of specific antibodies. Serial sectioning (2 μm thickness) was performed, and then sections were dewaxed and rehydrated in graded alcohol solutions. For heat-induced epitope retrieval, the sections were heated at 100 °C for 3 × 5 min using target retrieval solution pH 9 (DAKO). The nonspecific binding was blocked by pre-incubation with PBS for 20 min. An overnight incubation at 4 °C followed using accordingly the following antibodies: a. anti-Beclin 1 (1:200, #ab62557, Abcam), b. anti-SQSTM1/P62 (1:200, #ab91526, Abcam), c. anti-LC3A (1:50; #ap1805a, Absepta).

The slides were then washed with PBS and blocked using peroxidase (#SM801, DAKO) for 10 min. The slides were then incubated with EnVision Flex/HRP (#SM802, DAKO) for 30 min. After extensive washing with PBS (2 × 5 min), the color reaction was developed in 3,3′-diaminobenzidine (DAB) for 5 min and then again washed with PBS. The sections were then counterstained with hematoxylin (hematoxylin QS VECTOR Laboratories USA), dehydrated, and mounted. The expression of Beclin1 ≥ 10%, LCA ≥ 5%, and P62 ≥ 10% were defined as strong expressions of each specific antibody.

### 2.6. Immunoblotting

Snap-frozen liver tissues were homogenized at 4 °C using a plastic pestle in a sucrose-based lysis buffer (0.25 M sucrose, 25 mM Tris, pH 7.4) containing protease (#04693159001, Roche Diagnostics GmbH) and phosphatase inhibitors (#524629, Calbiochem) (60 μL lysis buffer per 10 mg of tissue). Tissue homogenates were incubated for 30 min on ice, and they were then centrifuged at 16,000× *g* for 20 min at 4 °C. The pellet contained the membrane fraction, which included autophagy-related vacuoles, mitochondria etc. (P), and the supernatant contained cytoplasmic soluble proteins (SP). [31] Total protein concentration was determined in the collected fractions by BCA protein assay (#23225, Thermo Scientific Pierce).

Samples (40 μg) were loaded on discontinuous SDS gels using 12% separating and 5% stacking gels. PVDF-PSQ membranes (Millipore Corp., Burlington, MA, USA) were employed for immunoblotting. Blots were horizontally cut into three parts, and the appropriate strips were incubated with p62, β-actin, and LC3A or LC3B antibodies. At first, the membranes were blocked with 5% non-fat dry milk in TBS-T (150 mM NaCl, 20 mM Tris, pH 7.5, 0.05% Tween-20) at room temperature for 60 min, and they were then hybridized overnight at 4 °C with primary anti-LC3A (1:700; #ap1805a, Absepta), anti-LC3B (1:1000; #LC3 5F10, Nanotools) or anti-SQSTM1/p62 (1:10,000, #ab109012, Abcam) antibodies. Each of these blots was also hybridized with an anti-β actin antibody (1:30,000, #NB600–501, Novus Biologicals). The PVDF membranes were then hybridized for 2 h at room temperature with the appropriate secondary antibody, goat polyclonal anti-rabbit IgG (H + L)-HRP (1:5000, #170–6515, Biorad) or goat polyclonal anti-mouse IgG (H + L)-HRP (1:5000, #170–6516, Biorad), and they were then developed in Clarity Western ECL Substrate (#1705060, Biorad). Blot images were captured using the iBright Imaging System, and densitometric analysis was performed using iBright Analysis software (Thermo Scientific Pierce). Nine samples were evaluated from each group.

### 2.7. Statistical Analysis

The collected data were analyzed statistically using the IBM SPSS Statistics for Windows, Version 24.0. Armonk, NY, USA: IBM Corp. Numerical variables are presented as mean ± standard deviation (SD) or as median (minimum–maximum), whereas categorical ones are presented using frequencies and valid percentages. A Student’s *t*-test and Analysis of Variance (ANOVA) were used for the statistical analysis of two and multiple sample means, respectively, when qualitative variables followed a normal distribution. Otherwise, Mann–Whitney or Kruskal–Wallis tests were utilized. The Kolmogorov–Smirnov test was implemented to check the normality of distribution. Odds ratios (OR) with their 95% confidence intervals (CI) were calculated for the nominal variables implementing the Haldane–Anscombe correction for frequencies equal to zero [32]. Regarding Western blot analysis, densitometric data were logarithmically transformed prior to any further analysis and are presented as geometric means with 95% CI. One-way ANOVA (Dunnett’s post hoc test) was employed for comparisons between the control or baseline group and treated groups, while a *t*-test was employed for comparisons between the *p* and the corresponding SP fractions. A *p*-value < 0.05 (two-tailed) was considered statistically significant.

## 3. Results

### 3.1. NAFLD Activity Score (NAS)

The control group exhibited the highest NAFLD Activity Score (NAS) among all study groups, with a mean value of 5.83 ± 1.27, compared to QUE 1 (4.75 ± 1.06), QUE 2 (5.17 ± 1.27), and BSL (3.73 ± 1.56) (*p* = 0.011). This result was primarily driven by the steatosis component of the NAS, with the control group demonstrating a steatosis level of 59.17 ± 25.39%, significantly higher than that of the other study groups (*p* < 0.001). The detailed NAS characteristics of each study group are presented in Table 1.

The control group demonstrated significantly higher rates of steatohepatitis (83.33%) compared to QUE 1 (33.3%) and QUE 2 (41.67%), with an odds ratio (OR) of 0.10 (95% CI: 0.01–0.69; *p* = 0.020) and 0.14 (95% CI: 0.02–0.96; *p* = 0.045), respectively, as illustrated in Figure 1. No significant difference was observed between the QUE 1 and QUE 2 groups. All mice in the BSL group had fatty liver, but none of them had progressed to steatohepatitis.

### 3.2. Immunohistochemistry

Table 2 and Table 3 present the expression of Beclin1, LCA, and P62 stratified by each study group. No significant difference was found between study groups regarding Beclin1 expression (*p*-value: 0.821). There was a statistically significant difference in the expression of LCA (*p*-value: <0.001) and P62 (*p*-value: 0.002) between the experimental groups.

All study groups showed a strong Beclin1 expression, and no statistically significant difference was found between them. Concerning LCA expression, QUE 1 had a statistically significant stronger expression compared to the control group (OR: 6.00; 95%CI: 1.02–35.38; *p*-value: 0.048), whereas QUE 2 also had a stronger expression than the control group, but it did not reach statistical significance (OR: 2.14; 95% CI: 0.38–12.20; *p*-value: 0.390). All BSL group mice showed a strong expression of LCA, which was statistically significant compared to the control group (OR: 62.43; 95% CI: 2.85–1365.60; *p*-value: 0.009). Regarding P62 expression, both QUE 1 and QUE 2 had a stronger expression compared to the control group, but it did not reach statistical significance with OR: 2.00; 95% CI: 0.38–10.41; *p*-value: 0.410 and OR: 1.43; 95% CI: 0.27–7.52; *p*-value: 0.674, respectively. All BSL animals had a strong expression of P62, which was statistically significant compared to the control group (OR: 43.44; 95% CI: 2.05–920.41; *p*-value: 0.016).

Most mice with steatohepatitis showed strong Beclin1 expression (68.42%), which was similar to the mice with a fatty liver (67.86%), with no statistically significant difference between the two groups (OR: 1.02; 95% CI: 0.29–3.59; *p*-value: 0.968). In contrast, 73.68% of the steatohepatitis group had weak LCA expression, which was significantly different from the fatty liver group, where 78.57% exhibited strong LCA expression (OR: 0.10; 95% CI: 0.02–0.38; *p*-value: <0.001). A statistically significant difference was also observed in P62 expression, with mice having steatohepatitis showing weaker expression compared to those with a fatty liver (73.68% vs. 25.00%) (OR: 0.12; 95% CI: 0.03–0.45; *p*-value: 0.002).

### 3.3. Immunoblotting

Subcellular fractionation enabled us to study whether p62, LC3A and LC3B were detected as autophagy-related vesicles (P fraction) or as soluble cytoplasmic forms (SP fraction). As shown in Figure 2, no significant differences in p62 expression were evident among groups in both fractions. P62 levels were significantly higher in the P fraction compared to the corresponding SP fraction in both QUE 1 (P: 1.30, 95% CI: 0.99–1.72 vs. SP: 0.88, 95% CI: 0.70–1.10; *p* < 0.05) and QUE 2 (P: 1.29, 95% CI: 1.05–1.58 vs. SP: 0.92, 95% CI: 0.72–1.16; *p* < 0.05) groups. A significant difference between P and SP fractions was also observed in the control group (P: 1.55, 95% CI: 1.30–1.86 vs. SP: 0.81, 95% CI: 0.58–1.14; *p* < 0.01), while no significant difference was noted in the baseline group (P: 1.46, 95% CI: 1.06–2.00 vs. SP: 1.16, 95% CI: 0.83–1.64; *p* > 0.05).

Concerning LC3A (Figure 3), LC3A-II was generally undetectable; thus, only LC3A-I was efficiently quantified by densitometry. A significant difference in LC3A-I present in P and SP fractions was detected in QUE 2 (P: 0.30, 95% CI: 0.24–0.36 vs. SP: 0.46, 95% CI: 0.36–0.60; *p* < 0.01), the control group (P: 0.26, 95% CI: 0.21–0.33 vs. SP: 0.44, 95% CI: 0.34–0.56; *p* < 0.01), and the baseline group (P: 0.26, 95% CI: 0.20–0.35 vs. SP: 0.38, 95% CI: 0.30–0.47; *p* < 0.05), but not in QUE 1 group (P: 0.26, 95% CI: 0.21–0.34 vs. SP: 0.39, 95% CI: 0.27–0.57; *p* > 0.05). Furthermore, LC3A-I expression did not differ significantly among groups in both fractions.

As illustrated in Figure 4, both LC3B-I and LC3B-II were detected in the fractions. LC3B-I levels differed significantly between fractions only in the baseline group (P: 0.09, 95% CI: 0.06–0.15 vs. SP: 0.17, 95% CI: 0.11–0.25; *p* < 0.05). LC3B-ΙΙ differed marginally in the QUE 2 (P: 1.37, 95% CI: 1.18–1.59 vs. SP: 1.06, 95% CI: 0.85–1.33; *p* = 0.05) and the control group (P: 1.52, 95% CI: 1.15–2.00 vs. SP: 1.09, 95% CI: 0.87–1.37; *p* = 0.051). No significant differences were found in LC3B-I or LC3B-II levels between P and SP fractions in the other groups, and there were no significant differences in LC3B-I or -II expression among groups, either.

Further subclassification of animals within each group, depending on the presence or absence of steatohepatitis, revealed that LC3A-I levels were higher in the P fraction of mice exhibiting steatohepatitis compared to mice exhibiting fatty liver disease in the CON group (*p* < 0.05). Apart from this, the expression levels of the membrane-bound or cytosolic forms of LC3A, LC3B and p62 were similar in all groups independently of the degree of liver injury. In the BSL group, mice exhibiting fatty liver disease demonstrated significantly higher levels of LC3A-I and LC3B-I in the SP than in the P fraction (*p* < 0.05) (Figure 4). Even though statistically insignificant due to the very low number of samples (only two mice demonstrated fatty liver), the latter was also noticed in the CON group. High LC3A-I levels in the SP fraction were also apparent in mice with a fatty liver belonging to the QUE 2 group (*p* < 0.05). Significantly higher p62 levels in the P fraction were noticed in mice with a fatty liver belonging to the QUE 1 group (*p* < 0.05). As far as steatohepatitis is concerned, mice belonging to the CON group had higher levels of p62 and LC3B-II in the P fraction (*p* < 0.01) and higher levels of LC3A-I in the SP fraction (*p* < 0.01). In general, due to, in some cases, the low number of samples or the large variance present within sub-groups, these results should be interpreted with caution.

## 4. Discussion

Non-alcoholic fatty liver disease (NAFLD) poses a major public health problem with increasing prevalence [1]. No definite pharmacotherapy is available against this disease, but natural polyphenols, including quercetin, have shown promising results with hepatoprotective action [15]. This study provides new insights into the role of quercetin in the management of NAFLD, particularly in relation to autophagy-related proteins Beclin1, p62, LC3A, and LC3B. The results of this experimental study suggest that quercetin exerts a protective effect on liver histology, with evidence of reduced steatosis and inflammation, which are key components of NAFLD.

Quercetin administration in the form of its water-soluble lyophilized complex with HP-β-CD (Que-HP-β-CD) was associated with a significant reduction in NAFLD Activity Score (NAS) in the treated groups compared to the control group. The observed higher inflammation score in the baseline group compared to the control group may reflect acute inflammatory responses characteristic of early-stage NAFLD [33]. As NAFLD progresses, the immune system adapts, leading to a reduction in inflammatory markers during later stages [33]. This pattern aligns with findings that show a decrease in resident Kupffer cells and their replacement by monocyte-derived macrophages as the disease advances, resulting in altered inflammatory profiles. The NAS, which includes the assessment of steatosis, lobular inflammation, and hepatocellular ballooning, is a reliable measure of NAFLD severity. The control group, which continued on a high-fat diet without quercetin intervention, exhibited the highest NAS, indicating severe steatosis and a higher prevalence of steatohepatitis. Conversely, the groups treated with quercetin (QUE 1 and QUE 2) showed significantly lower NAS, particularly in the steatosis component, suggesting that quercetin effectively mitigates the progression of hepatic lipid accumulation and inflammation. These findings are consistent with previous studies that have demonstrated quercetin’s ability to reduce hepatic triglyceride accumulation and improve liver histology in NAFLD models [17].

The marked reduction in steatohepatitis prevalence observed in the quercetin-treated groups reinforces the idea that quercetin may help delay or prevent the progression of simple steatosis to more severe forms of NAFLD, such as non-alcoholic steatohepatitis (NASH). Li et al. demonstrated that quercetin alleviates liver inflammation and fibrosis in mice by inhibiting macrophage infiltration [34]. Similarly, Macrolin et al. reported an improvement in NASH, showing that quercetin attenuates several pro-fibrotic and pro-inflammatory pathways in mice [35]. He et al. explored the effects of the traditional Chinese Shenqi pill, with quercetin as its active compound, in a NASH rat model and found that liver injury and lipid metabolism significantly improved after its administration, primarily through the suppression of pro-inflammatory cytokines (IL-6, TNF-α) [36]. Our study evaluated the therapeutic effects of quercetin at two doses (10 mg/kg and 50 mg/kg) on NAFLD, but the findings did not demonstrate a clear dose-dependent relationship. Although the use of cyclodextrin complexes improved quercetin solubility, its bioavailability might plateau beyond a certain dose, limiting observable differences between the treatment groups.

Furthermore, its clinical application is restricted by poor water solubility, which greatly limits its bioavailability following oral administration. To address this challenge, complexation with natural cyclodextrins (CDs) and their derivatives, particularly hydroxypropyl-β-cyclodextrin (HP-β-CD) and methyl-β-cyclodextrin (Me-β-CD), has emerged as an effective strategy [27,28]. The structure of cyclodextrins comprises a hydrophilic outer surface and a hydrophobic inner cavity, which allows them to encapsulate lipophilic molecules or non-polar groups, thereby enhancing their aqueous solubility through the formation of inclusion complexes [37]. The improvement of quercetin’s aqueous solubility by the formation of quercetin–CD complexes has shown promise in improving the pharmacokinetic properties of the compound, making it a more viable candidate for therapeutic use [38].

Autophagy is a crucial process for maintaining cellular homeostasis and has been implicated in the pathogenesis of NAFLD [19]. Current evidence suggests that autophagy is downregulated in fatty liver [39]. Autophagosomes accumulate in hepatic cells, mainly due to the blocking of autophagosome and lysosome fusion [39]. This downregulation of the autophagy process leads to impaired lipophagy and, thus, further increases lipid accumulation [40]. In combination with excessive lipid intake, fat accumulation leads to the blockage of autophagic reflux and leads to hepatic steatosis. This vicious cycle of fat accumulation (due to excessive intake) and impairments in autophagic flux results in the development of hepatic steatosis [19]. Zhao et al. suggested that hepatic steatosis leads to a reduction of autophagy proteins, such as LC3 and Beclin-1 [41]. Quercetin has been shown to activate AMP-activated protein kinase (AMPK) while inhibiting the mechanistic target of rapamycin (mTOR), mechanisms that collectively enhance autophagy and reduce hepatic lipid accumulation in NAFLD models. Gnoni et al. highlighted that quercetin inhibits de novo fatty acid synthesis in hepatocytes through the ACACA/AMPK/PP2A axis, effectively decreasing lipid accumulation [42]. Similarly, Zhang et al. observed that AMPK/mTOR-mediated autophagy is impaired in high-fat-diet-fed mice, and targeting this pathway could mitigate hepatotoxicity [43]. Marcondes-de-Castro et al. highlighted the critical role of AMPK and mTOR pathways in maintaining hepatic homeostasis, proposing that targeting these mechanisms may offer therapeutic benefits for NAFLD management [44]. These findings underscore the importance of the AMPK/mTOR pathway in NAFLD progression and suggest that quercetin’s modulation of this pathway may offer therapeutic benefits. Future research should focus on elucidating the detailed mechanisms by which quercetin influences these signaling pathways to better understand its potential role in NAFLD treatment.

Beclin1 is a key initiator of autophagy, a process essential for cellular homeostasis, particularly in the liver, where it plays a crucial role in lipid metabolism and the prevention of hepatocyte degeneration [45,46]. In our study, Beclin1 expression was found to be consistently strong across all experimental groups, suggesting that the initiation of autophagy via Beclin1 is preserved in NAFLD, regardless of the disease stage or quercetin treatment. The preservation of Beclin1 expression in NAFLD models, including those with more severe steatohepatitis, indicates that the initiation of autophagy is not necessarily impaired during the disease’s progression. Instead, the impairment may occur downstream, affecting the maturation and fusion of autophagosomes with lysosomes. These findings are consistent with previous research indicating that although autophagy initiation is preserved in NAFLD, downstream processes may be impaired, resulting in autophagosome accumulation and disrupted degradation of intracellular substrates [47]. On the contrary, Menk et al. suggested that a downregulated Beclin1 expression may result in liver injury, but this study concerned mice with liver injury after chronic alcohol consumption. [48]

P62, also referred to as sequestosome 1 (SQSTM1), is a versatile protein that plays a crucial role in selective autophagy by targeting ubiquitinated proteins for degradation [49]. It also plays a role in signaling pathways related to inflammation and oxidative stress, both of which are critical in the pathogenesis of NAFLD [21,40]. Accumulation of p62 often indicates impaired autophagic flux [50]. Impaired flux results in the buildup of cellular waste, contributing to inflammation and oxidative stress, which drive NAFLD progression. 

Furthermore, p62 has been shown to aggregate within hepatocytes during the advanced stages of NAFLD, suggesting its association with disease severity [50]. In the present study, immunohistochemical analysis revealed that p62 expression was higher in quercetin-treated groups compared to the control group, although these differences did not reach statistical significance. Following immunoblotting analysis of subcellular fractions, it was evident that even though p62 expression was similar in all groups, the presence of p62 inclusion bodies was increased in mice receiving HFD diet for 16 weeks, with or without QUE treatment. Since most mice in the CON group (HFD diet for 16 weeks) exhibited steatohepatitis, while quercetin-treated mice mostly exhibited fatty liver, the accumulation of p62 in the membrane fraction could indicate its implication in other cellular processes apart from autophagy. In addition, it could be acceptable to suggest that the presence of quercetin seems to affect p62 accumulation, and furthermore, it is possible that p62’s participation in other cellular processes might be associated with the protective effect of quercetin against steatohepatitis. Quercetin promotes the activation of Nuclear factor erythroid 2-related factor 2 (Nrf2), a master regulator of cellular defense mechanisms, which enhances antioxidant defenses, mitigates oxidative stress, stimulates mitochondrial biogenesis, and suppresses fatty acid uptake and de novo synthesis [51,52,53]. Under oxidative stress conditions, p62 binds to Kelch-like ECH-associated protein 1 (Keap1), a repressor of Nrf2, effectively sequestering it and preventing Nrf2 degradation. This stabilization allows Nrf2 to translocate into the nucleus, where it promotes the transcription of antioxidant and cytoprotective genes, such as those encoding for glutathione synthesis and detoxification enzymes. By doing so, the p62-Nrf2 axis not only reduces oxidative damage but also supports mitochondrial functionality and overall liver homeostasis. This pathway thus serves as a crucial mechanism in mitigating the early stages of NAFLD progression, demonstrating its potential as a promising therapeutic target [54]. Under stress conditions, p62 binds Keap1, which normally represses NRF2, preventing its degradation [55]. This stabilization allows NRF2 to translocate into the nucleus, where it activates genes that promote antioxidant defenses. By reducing oxidative damage and supporting cellular homeostasis, the p62-NRF2 pathway may play a protective role in NAFLD progression, showing its potential as a therapeutic target [56]. However, as the disease progresses, as seen in the control group, this autophagic capacity may be overwhelmed, leading to p62 dysregulation and further liver damage [25]. Tan et al. have also suggested that dysregulation of p62 activity is strongly associated with the development and pathogenesis of NAFLD. They also reported that an increase of p62 bodies leads to liver disease and could be potentially used as a biomarker for NASH and hepatocellular carcinoma [25]. Further studies by Carotti et al. and Zatloukal et al. showed that high levels of p62 in NAFLD patients lead to disease progression to fibrosis and steatohepatitis [57,58]. Given its multifaceted roles in autophagy, inflammation, and oxidative stress, p62 represents a promising therapeutic target. Quercetin shows a therapeutic effect against NAFLD by decreasing P62 accumulation, thus leading to decreased lipid accumulation [59].

LC3A and LC3B are critical components of the autophagic machinery, involved in the elongation of autophagosomal membranes and the formation of autophagosomes [22]. Their expression levels and processing (conversion from LC3-I to LC3-II) are commonly used as markers of autophagic activity in combination with p62 expression [21,22]. In our study, according to immunohistochemical analysis, the baseline group exhibited an upregulation of LC3A expression, while the control group exhibited a reduced expression. According to Western blotting analysis, only LC3A-I, mainly in its cytosolic form, was detected in all samples. The absence of LC3A-II in non-alcoholic fatty liver disease (NAFLD) may result from inhibited lipidation of LC3A-I, preventing its conversion to LC3A-II. Alternatively, LC3A-II might play a less significant role in autophagy within NAFLD contexts. LC3 family proteins exhibit distinct subcellular localizations and functions, suggesting non-redundant roles in autophagy processes [60]. Specifically, LC3A has been observed to localize predominantly in the perinuclear region and within the nucleus, whereas LC3B is distributed throughout the cytoplasm. This differential localization implies that LC3A and LC3B may be involved in the autophagic degradation of distinct cellular components [60].

Upregulation of cytosolic LC3A-I was evident in mice with a fatty liver belonging to the baseline and high-dose quercetin-treated groups, as well as in mice with NASH in the control group. LC3B-I was also detected mainly in the cytoplasm of the baseline group of mice with fatty liver. Mice with a fatty liver belonging to the control group also demonstrated high cytosolic LC3A-I and LC3B-I levels, but due to the very small number of animals, the results should be interpreted with caution. In addition, animals with NASH in the control group demonstrated significantly higher LC3A-I levels in the membrane fraction compared to mice with fatty liver in the same group. The differences in inflammation scores, as well as LCA and p62 levels, between the baseline and control groups, illustrate the dynamic nature of autophagic activity and immune responses during NAFLD progression. The baseline group likely represents an earlier stage of the disease where autophagy is actively mitigating metabolic and inflammatory stress. In contrast, the control group reflects later-stage NAFLD, where autophagic capacity may become impaired, resulting in the accumulation of dysfunctional proteins and organelles. These findings highlight the importance of autophagic regulation in NAFLD progression and underscore the complexity of interpreting protein expression changes at different disease stages. 

Runwal et al. suggested that in case of LC3-II absence or impairment, LC3-I forms puncta with p62, resulting in p62 accumulation [61]. On the other hand, the activation of autophagy causes the lipidation of LC3-I to LC3-II, recruiting the ubiquitinated p62 and then leading them to autophagosomes for degradation [61]. NASH was associated with elevated levels of cytosolic LC3A-I, p62 inclusion bodies accumulation in conjunction with membrane-bound LC3B-II. The accumulation of p62 and LC3B-II possibly indicates the formation of autophagosomes but the lack of lysosomal degradation [62]. Furthermore, impaired lysosomal acidification in settled NAFLD leads to impaired autophagic degradation and clearance [63].

Interestingly, the lack of significant differences in LC3A and LC3B expression in the quercetin low-dose group may be indicative that quercetin’s effects on autophagy may be dose-dependent. Higher doses of quercetin may be necessary to achieve significant enhancement of autophagic activity [64]. Rodriguez et al. found that an increase in the LC3-II/LC3-I ratio is associated with disease progression to steatohepatitis [21]. Willy et al. also showed that NAFLD induces autophagy through LC3B enhancement, and autophagy impairment leads to disease progression to steatohepatitis [22].

As expected, our study is subject to certain limitations. The study was carried out in an experimental mouse model, and further research is needed to confirm these findings in human subjects. The relatively small number of animals included in each experimental group may have limited the statistical power to detect subtle dose-dependent differences, a recognized limitation in preclinical studies. Additionally, the precise molecular mechanisms by which quercetin influences autophagy-related proteins remain to be fully elucidated. An additional key limitation of this study was the absence of biochemical analysis of liver function tests, which could have complemented the histological findings. Future studies should integrate biochemical, histological, and molecular assessments to provide a more comprehensive understanding of NAFLD progression and treatment effects. Additionally, future research should focus on identifying the upstream regulators of autophagy influenced by quercetin and determining whether these effects persist across various stages of NAFLD.

## 5. Conclusions

In conclusion, this study provides compelling evidence that quercetin, when administered orally in the form of Que-HP-β-CD lyophilized product, can attenuate the severity of NAFLD by modulating key autophagy-related pathways. The consistent expression of Beclin1 across all groups suggests that quercetin’s effects are more pronounced in enhancing downstream autophagic processes rather than initiating autophagy. The modulation of p62 and LC3A, and LC3B proteins by quercetin indicates an improvement in autophagic flux, which is crucial for the degradation of lipid droplets and the maintenance of liver homeostasis.

Quercetin’s observed effects on autophagy modulation and reduction of liver injury markers in this study suggest its potential as a therapeutic candidate for human NAFLD. These findings are promising, but translating results from preclinical models to human applications requires careful consideration. Physiological and metabolic differences between mice and humans may influence the efficacy and safety of quercetin. Future research should prioritize randomized controlled trials to validate these results in humans. These trials should aim to establish optimal dosing strategies, assess long-term safety profiles, and explore the compound’s effectiveness across diverse patient populations. Additionally, understanding quercetin’s interactions with other therapeutic agents and its pharmacokinetics in human systems will be crucial for its clinical translation. Such studies will provide a clearer understanding of quercetin’s potential as a therapeutic agent for NAFLD and facilitate its development into a viable treatment option.

## Figures and Tables

**Figure 1 nutrients-16-04282-f001:**
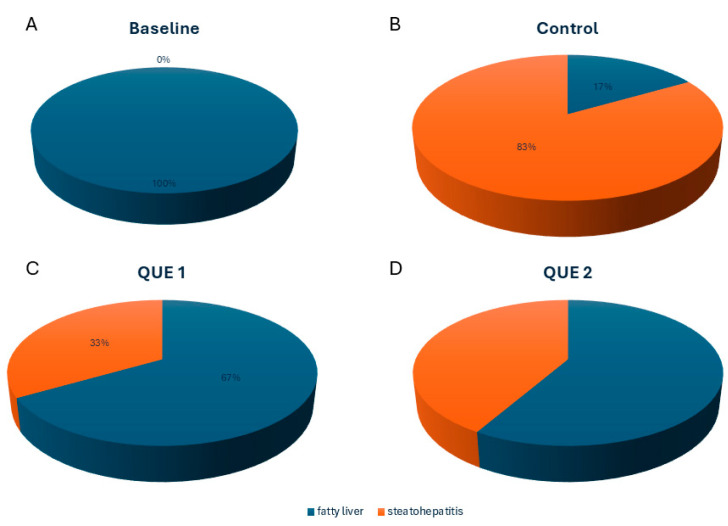
Steatohepatitis prevalence was stratified by each experimental group ((**A**) Baseline, (**B**) Control, (**C**) QUE 1, (**D**) QUE 2). Orange segments indicate the prevalence of steatohepatitis, while blue segments represent the prevalence of fatty liver. QUE 1—low-dose quercetin, QUE 2—high-dose quercetin.

**Figure 2 nutrients-16-04282-f002:**
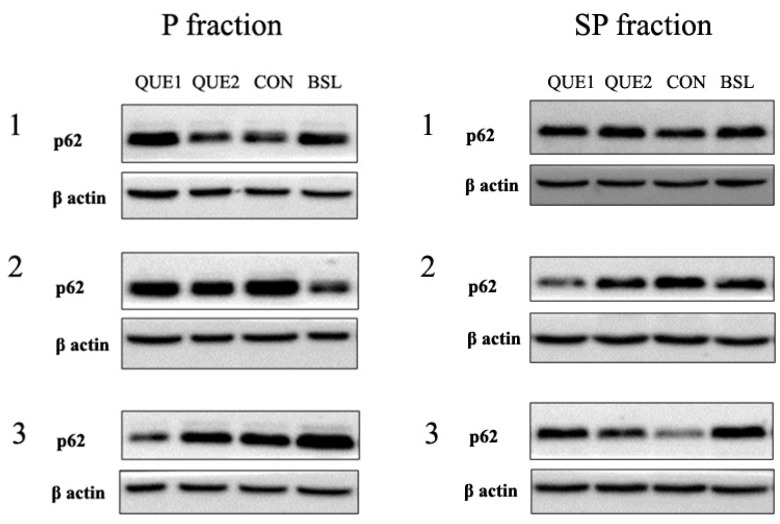
Representative immunoblotting images and densitometric analysis of p62 expression. The P fraction contains autophagy-related vesicles, and the SP fraction contains cytoplasmic soluble proteins. The bold horizontal lines in the graphs represent the geometric means. Where #, *p* < 0.05 vs. the corresponding P fraction. QUE 1—low-dose quercetin, QUE 2—high-dose quercetin, CON—control, BSL—baseline.

**Figure 3 nutrients-16-04282-f003:**
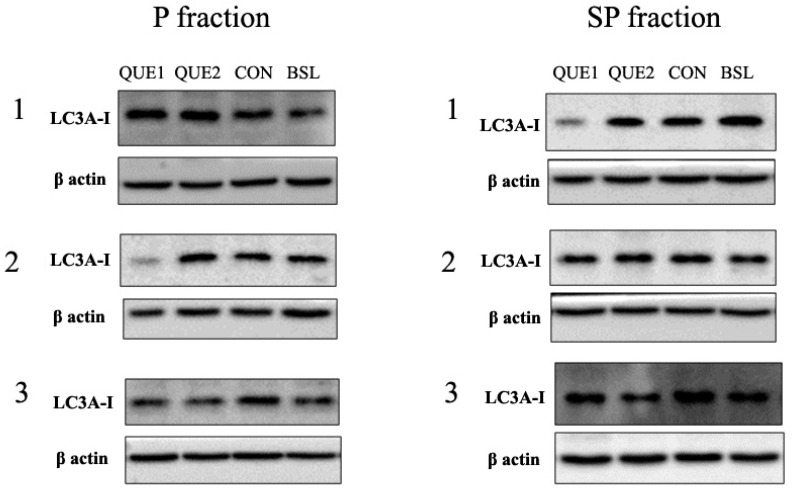
Representative immunoblotting images and densitometric analysis of LC3A-I expression. In general, LC3A-II was undetectable. The P fraction contains autophagy-related vesicles, and the SP fraction contains cytoplasmic soluble proteins. The bold horizontal lines in the graphs represent the geometric means. Where #, *p* < 0.05 vs. the corresponding P fraction. QUE 1—low-dose quercetin, QUE 2—high-dose quercetin, CON—control, BSL—baseline.

**Figure 4 nutrients-16-04282-f004:**
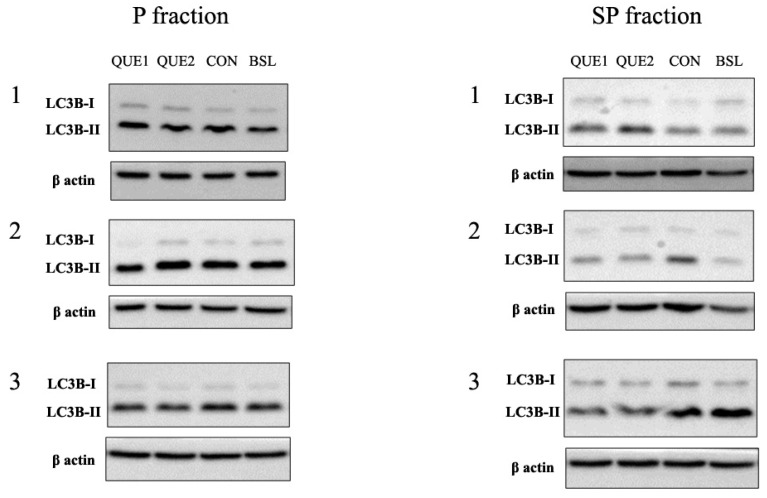
Representative immunoblotting images and densitometric analysis of LC3B-I and -II expression. The P fraction contains autophagy-related vesicles, and the SP fraction contains cytoplasmic soluble proteins. The bold horizontal lines in the graphs represent the geometric means. Where #, *p* ≤ 0.05 vs. the corresponding P fraction. QUE 1—low-dose quercetin, QUE 2—high-dose quercetin, CON—control, BSL—baseline.

**Table 1 nutrients-16-04282-t001:** NAFLD Activity score (NAS) and its components were stratified by each experimental group.

GROUP	N	NAS	STEATOSIS (%)	STEATOSIS SCORE	INFLAMMATION SCORE	BALLOONING SCORE
BASELINE	11	3.73 ± 1.56	9.09 ± 11.9	0.55 ± 0.69	1.64 ± 0.67	1.55 ± 0.69
QUE 1	12	4.75 ± 1.06	35.33 ± 20.96	1.58 ± 0.79	1.17 ± 0.83	2.00 ± 0.00
QUE 2	12	5.17 ± 1.27	38.08 ± 27.84	1.58 ± 0.9	1.67 ± 0.49	1.92 ± 0.29
CONTROL	12	5.83 ± 1.27	59.17 ± 25.39	2.33 ± 0.78	1.50 ± 1.00	2.00 ± 0.00

QUE 1—low-dose quercetin, QUE 2—high-dose quercetin.

**Table 2 nutrients-16-04282-t002:** Beclin1, LCA, and P62 expression (%) stratified by each experimental group.

GROUP	N	Beclin1 (%)	LCA (%)	P62 (%)
BASELINE	11	11.82 ± 11.46	17.18 ± 10.34	25.45 ± 8.20
QUE 1	12	8.75 ± 4.83	6.25 ± 14.16	8.33 ± 11.15
QUE 2	12	12.92 ± 8.65	3.33 ± 2.46	9.17 ± 11.65
CONTROL	12	10.00 ± 5.64	1.83 ± 3.49	6.67 ± 11.55

QUE 1—low-dose quercetin, QUE 2—high-dose quercetin.

**Table 3 nutrients-16-04282-t003:** Beclin1, LCA, and P62 expression stratified by each experimental group.

GROUP	N	BECLIN1 N (%)		LCA N (%)		P62 N (%)	
		Weak	Strong	Weak	Strong	Weak	Strong
BASELINE	11	4 (36.36%)	7 (63.64%)	0 (0.00%)	11 (100.0%)	0 (0.00%)	11 (100.0%)
QUE 1	12	4 (33.33%)	8 (66.67%)	4 (33.33%)	8 (66.67%)	6 (50.00%)	6 (50.00%)
QUE 2	12	4 (33.33%)	8 (66.67%)	7 (58.33%)	5 (41.67%)	7 (58.33%)	5 (41.67%)
CONTROL	12	3 (25.00%)	9 (75.00%)	9 (75.00%)	3 (25.00%)	8 (66.67%)	4 (33.33%)

QUE 1—low-dose quercetin, QUE 2—high-dose quercetin.

## Data Availability

Data are contained within the article.

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
