# Peer review of "The Effect of Quercetin on Non-Alcoholic Fatty Liver Disease (NAFLD) and the Role of Beclin1, P62, and LC3: An Experimental Study"

_nutrients, 2024, doi:10.3390/nu16244282_

Round 1

Reviewer 1 Report

Comments and Suggestions for Authors

Strengths:

  • Relevant Topic: NAFLD is a growing health concern, and exploring natural compounds like quercetin as potential therapeutic options is important.
  • Well-Designed Study: The study design is generally sound, with a clear experimental setup, including a baseline group, control group, and two quercetin treatment groups.
  • Comprehensive Analysis: The researchers used a combination of methods, including NAS scoring, immunohistochemistry, and immunoblotting, to assess the effects of quercetin on NAFLD and autophagy.
  • Interesting Findings: The study provides evidence for the potential beneficial effects of quercetin in reducing NAFLD severity and modulating autophagy-related proteins.

Areas for Improvement:

  • Clarify Quercetin Formulation: While the manuscript mentions the use of a Que-HP-β-CD lyophilized product, it could benefit from a more detailed description of its preparation and characterization. This would help ensure reproducibility and allow for better comparison with other studies.
  • Discuss Limitations of NAS: While NAS is a widely used scoring system, it has limitations. The authors should acknowledge these limitations and consider including other measures of liver injury, such as serum ALT and AST levels, for a more comprehensive assessment.
  • Further Explore Autophagy Mechanisms: The study suggests that quercetin modulates autophagy, but the exact mechanisms remain unclear. Further investigation into the specific pathways involved, such as the role of mTOR and AMPK, would strengthen the conclusions.
  • Expand on the Role of p62: The manuscript highlights the complex role of p62 in autophagy and other cellular processes. A more in-depth discussion of its potential implications in NAFLD pathogenesis and as a therapeutic target would be valuable.
  • Address Discrepancies in LC3A-II: The authors note that LC3A-II was undetectable in their immunoblotting analysis. This finding should be discussed in more detail, considering its potential implications for the interpretation of autophagic activity.
  • Clarify Dose-Dependent Effects: The study used two different doses of quercetin, but the results do not clearly demonstrate a dose-dependent effect. This aspect should be addressed and discussed in the context of potential clinical applications.
  • Strengthen Clinical Relevance: While the study provides promising results in a mouse model, its clinical relevance needs to be emphasized. The authors should discuss the potential implications for human NAFLD and suggest future research directions, including clinical trials.

Specific Comments:

  • Introduction: Provide a more concise overview of NAFLD, focusing on the key aspects relevant to the study.
  • Methods: Specify the composition of the high-fat diet used to induce NAFLD.
  • Results: Consider presenting the immunohistochemistry and immunoblotting data in a more visually appealing format, such as bar graphs or scatter plots.
  • Discussion: Relate the findings to existing literature on quercetin and NAFLD, highlighting the novelty and significance of the study.
  • Conclusion: Provide more specific recommendations for future research and potential clinical applications of quercetin in NAFLD.

Author Response

Thank you for your valuable feedback, which will significantly contribute to the improvement of our manuscript.
Comment 1: Clarify Quercetin Formulation: While the manuscript mentions the use of a Que-HP-β-CD lyophilized product, it could benefit from a more detailed description of its preparation and characterization. This would help ensure reproducibility and allow for better comparison with other studies.
Response 1: Thank you for your suggestion. In terms of preparation of the Que-HP-β-CD lyophilized product, we have included all the steps performed, as well as the exact amounts used in the procedure (Materials and Methods, Section 2.3). For the characterization of the Que-HP-β-CD lyophilized product, a brief description of the results of the previous studies from DSC, XRD, SEM, solubility and dissolution characterization is included in the revised manuscript [section 2.3]. 
Comment 2: Discuss Limitations of NAS: While NAS is a widely used scoring system, it has limitations. The authors should acknowledge these limitations and consider including other measures of liver injury, such as serum ALT and AST levels, for a more comprehensive assessment.
Response 2: We appreciate the suggestion to include biochemical measures such as serum ALT and AST levels to provide a more comprehensive assessment of liver injury. Unfortunately, blood samples were not collected in this study, as the focus was primarily on histological and molecular analyses. We acknowledge this limitation in the revised manuscript.
Comment 3: Further Explore Autophagy Mechanisms: The study suggests that quercetin modulates autophagy, but the exact mechanisms remain unclear. Further investigation into the specific pathways involved, such as the role of mTOR and AMPK, would strengthen the conclusions.
Response 3: Thank you for your comment. We expanded the Discussion section to include potential roles of mTOR and AMPK pathways, emphasizing their interaction with quercetin.
Comment 4: Expand on the Role of p62: The manuscript highlights the complex role of p62 in autophagy and other cellular processes. A more in-depth discussion of its potential implications in NAFLD pathogenesis and as a therapeutic target would be valuable.
Response 4: Thank you for your feedback. We expanded discussion on p62's implications in NAFLD pathogenesis and its role as a therapeutic target, including detailed references to NRF2 activation.
Comment 5: Address Discrepancies in LC3A-II: The authors note that LC3A-II was undetectable in their immunoblotting analysis. This finding should be discussed in more detail, considering its potential implications for the interpretation of autophagic activity.
Response 5: Thank you for your input. We further clarified this finding in the respective Discussion section.
Comment 6: Clarify Dose-Dependent Effects: The study used two different doses of quercetin, but the results do not clearly demonstrate a dose-dependent effect. This aspect should be addressed and discussed in the context of potential clinical applications.
Response 6: Thank you for your comment. While our results did not show a clear dose-dependent trend, this could be attributed to the specific experimental conditions or threshold effects. We added a relevant section to the Discussion.
Comment 7: Strengthen Clinical Relevance: While the study provides promising results in a mouse model, its clinical relevance needs to be emphasized. The authors should discuss the potential implications for human NAFLD and suggest future research directions, including clinical trials.
Response 7: Thank you for highlighting this important aspect. In the revised manuscript, we have expanded the Conclusion section to include a paragraph on the clinical relevance of our findings.
Comment 8: Provide a more concise overview of NAFLD, focusing on the key aspects relevant to the study.
Response 8: Thank you for your suggestion. In the revised manuscript we updated the Introduction to provide a more concise and focused overview of NAFLD
Comment 9: Specify the composition of the high-fat diet used to induce NAFLD.
Response 9: Thank you for your comment. We add high-fat diet composition in the Methods Section.
Comment 10: Consider presenting the immunohistochemistry and immunoblotting data in a more visually appealing format, such as bar graphs or scatter plots.
Response 10: We appreciate the reviewer’s suggestion to present the immunohistochemistry and immunoblotting data in a more visually appealing format. However, the software and tools used for data acquisition and analysis inherently limit further improvements in display formatting.
Comment 11: Relate the findings to existing literature on quercetin and NAFLD, highlighting the novelty and significance of the study.
Response 11: Thank you for your input. We added a relevant statement in the Discussion.
Comment 12: Provide more specific recommendations for future research and potential clinical applications of quercetin in NAFLD.
Response 12: Thank you for your comment. We added a relevant paragraph in the Conclusions combined with your Comment 7.

Reviewer 2 Report

Comments and Suggestions for Authors

Polyphenols are compounds that are commonly found in nature, and their biological activity has been extensively described in the literature. Quercetin is also one of these compounds. Despite this, scientific research is still being conducted to find new therapeutic applications for representatives of this class of compounds.

The introduction to the article was written based on the latest literature and provides the reader with a good introduction to the subject of the presented research.

The studies conducted on mice were logically planned to achieve the intended effect and were supported by solid literature support. The authors presented the obtained results in a very clear and unambiguous way, and the conclusions drawn from them are completely justified. The presented work seems to be an interesting introduction encouraging further research.

Author Response

Comment 1: Polyphenols are compounds that are commonly found in nature, and their biological activity has been extensively described in the literature. Quercetin is also one of these compounds. Despite this, scientific research is still being conducted to find new therapeutic applications for representatives of this class of compounds.
The introduction to the article was written based on the latest literature and provides the reader with a good introduction to the subject of the presented research.
The studies conducted on mice were logically planned to achieve the intended effect and were supported by solid literature support. The authors presented the obtained results in a very clear and unambiguous way, and the conclusions drawn from them are completely justified. The presented work seems to be an interesting introduction encouraging further research.
Response 1: We sincerely thank the reviewer for their positive feedback and for recognizing the strengths of our study

Reviewer 3 Report

Comments and Suggestions for Authors

The manuscript investigates the protective effects of quercetin (a natural polyphenol) in NAFLD, using an in vivo model in mice. The research is interesting, providing new data which can contribute to the clarification of the molecular mechanisms of drugs potentially useful in NAFLD. However, a few issues need to be corrected or better explained in the manuscript.

1. In Table 1, why the inflammation score of the baseline group was higher than the control group, which received an additional 4 week administration of high-fat diet??? Please explain.

2. Figure 1 is difficult to comprehend, you need to calculate the ratio of 10 out of 12 to give you 83%. I recommend to change the figure into a "pie graph", directly with the percentages, it would be much more easy to read.

3. In Table 2, in table head you should mention that these are percentages (%). And again, why the LCA and P62 level in baseline group (receiving 12 weeks of high-fat diet) is so different from control group (receiving 16 weeks of HFD???

4. In all the Tables, you have put the line "Total". What is the point of summing up the data from all the groups and making an arithmetical mean???. The tables should give you an image of the data in the treated groups compared with the control group, why summing up all the groups together???

Author Response

Comment 1: The manuscript investigates the protective effects of quercetin (a natural polyphenol) in NAFLD, using an in vivo model in mice. The research is interesting, providing new data which can contribute to the clarification of the molecular mechanisms of drugs potentially useful in NAFLD. However, a few issues need to be corrected or better explained in the manuscript.
In Table 1, why the inflammation score of the baseline group was higher than the control group, which received an additional 4 week administration of high-fat diet??? Please explain.
Response 1: We thank the reviewer for their thoughtful feedback and constructive suggestions, which have helped improve the quality of our manuscript. 
We appreciate your observation regarding the baseline group having a higher inflammation score compared to the control group. This discrepancy likely reflects the resolution of early inflammatory responses as NAFLD progresses to later stages in the control group. This explanation has been included in the Results and Discussion sections.
Comment 2: Figure 1 is difficult to comprehend, you need to calculate the ratio of 10 out of 12 to give you 83%. I recommend to change the figure into a "pie graph", directly with the percentages, it would be much more easy to read.
Response 2: Thank you for your comment. We changed Figure 1 to pie graph to better visualize our results.
Comment 3: In Table 2, in table head you should mention that these are percentages (%). And again, why the LCA and P62 level in baseline group (receiving 12 weeks of high-fat diet) is so different from control group (receiving 16 weeks of HFD???
Response 3: Thank you for your insightful comments. We have updated the table header in Table 2 to clarify that the values represent percentages (%). Regarding the differences in LCA and p62 levels between the baseline and control groups, we believe these reflect dynamic changes in autophagic activity during NAFLD progression. We added a relevant statement in the Discussion section.
Comment 4: In all the Tables, you have put the line "Total". What is the point of summing up the data from all the groups and making an arithmetical mean???. The tables should give you an image of the data in the treated groups compared with the control group, why summing up all the groups together???
Response 4: Thank you for your input. We removed "Total" rows from all tables to focus on group-wise comparisons.

Round 2

Reviewer 3 Report

Comments and Suggestions for Authors

I am glad that the authors have improved the quality of their manuscript, according to reviewers' suggestions, therefore I recommend the publication of the revised version of their manuscript.